# Numerical Study of Thin-Walled Polymer Composite Part Quality When Manufactured Using Vacuum Infusion with Various External Pressure Controls

**DOI:** 10.3390/polym16050654

**Published:** 2024-02-28

**Authors:** Sergey Shevtsov, Shun Hsyung Chang, Igor Zhilyaev, Boon Xian Chai, Natalia Snezhina

**Affiliations:** 1Department of Composite Materials and Structures, Southern Center of Russian Academy of Science, 344006 Rostov on Don, Russia; 2Department of Microelectronic Engineering, National Kaohsiung University of Science and Technology, Kaohsiung City 82445, Taiwan; 39T Labs, 8048 Zurich, Switzerland; igor.zhilyaev@9tlabs.com; 4Aerostructures Innovation Research (AIR) Hub, Swinburne University of Technology, Hawthorn, VIC 3122, Australia; bchai@swin.edu.au; 5Department of Aircraft Engineering, Don State Technical University, 344000 Rostov on Don, Russia; snezhina_nataly@mail.ru

**Keywords:** polymeric composites, thin-walled composite structures, vacuum infusion technology, controlled pressures application, finite element modeling and optimization

## Abstract

The article presents the results of modeling various modes of vacuum infusion molding of thin-walled polymer-composite structures of arbitrary geometry. The small thickness of the manufactured structures and the fixation of their back surface on the rigid surface of the mold made it possible to significantly simplify the process model, which takes into account the propagation of a thermosetting resin with changing rheology in a compressible porous preform of complex 3D geometry, as well as changes in boundary conditions at the injection and vacuum ports during the post-infusion molding stage. In the four modes of vacuum-infusion molding studied at the post-infusion stage, the start time, duration and magnitude of additional pressure on the open surface of the preform and in its vacuum port, as well as the state of the injection gates, were controlled (open–closed). The target parameters of the processes were the magnitude and uniformity of the distribution of the fiber volume fraction, wall thickness, filling of the preform with resin and the duration of the process. A comparative analysis of the results obtained made it possible to identify the most promising process modes and determine ways to eliminate undesirable situations that worsen the quality of manufactured composite structures. The abilities of the developed simulation tool, demonstrated by its application to the molding process of a thin-walled aircraft structure, allow one to reasonably select a process control strategy to obtain the best achievable quality objectives.

## 1. Introduction

Increasing demands on the strength, stability and aerodynamic precision of many thin-walled aircraft structures made from polymer composite materials [1,2,3] have prompted a large amount of research into design methods [4,5,6,7,8,9] and technologies [10,11,12,13] capable of meeting these requirements at satisfactory performance and cost. An analysis of the results of these studies shows that in practice, meeting these often conflicting requirements with restrictions on mass, properties of available components and technologies is a rather complex task that can be solved by making expert decisions.

For example, increasing the stiffness of the polymer matrix by adding reinforcing particles to it, optimizing the thickness [14] and the fiber volume fraction [15] distributions in the preform, and the angles and order of reinforcing layers winding/laying [7,8,9], depending on the dynamic nature and area of application of the maximum loads, will improve the strength characteristics of the composite structure. But even if the weight requirements are met, they significantly complicate and increase the cost of production. In addition, the implementation of the described methods can be difficult when forming thin-walled composite structures (see Figure 1), most frequently in open molds (see Figure 2) with the injection of liquid resin into a dry preform using a vacuum.

These difficulties are associated primarily with the complexity of propagation of the liquid resin front in a three-dimensional preform of complex shape, changes in space and time of the pore pressure gradient, resin viscosity and anisotropic permeability of the preform [12,13]. To achieve acceptable quality and productivity indicators for such processes of molding composite structures as RTM, VAP and its varieties, it is necessary to solve the problem of finding optimal control. Its controllable parameters typically include the location of the resin inlet and outlet ports, the pressures at these ports, and temperature. The solution to such a problem is carried out either by trial and error, or using computer modeling tools, which have become increasingly widespread in recent years [10,11,12,13,14,15,16].

However, an alternative way to avoid the disadvantages and difficulties of obtaining reliable results in the production of critical composite structures by such methods is to technically modify them, most often by introducing additional controllable parameters. Such technical solutions include the so-called pulsed vacuum infusion process, the use of a post-infusion dwell for about 2 h at high temperature [17,18,19,20,21,22], which allows reduced porosity and increased strength to be obtained. Nevertheless, the most effective and fairly widespread solution to the problem of improving the vacuum infusion process seems to be through the technology of controlled post-infusion external pressure, proposed and studied on samples in [23,24]. This modification involves applying varying pressure to the exposed surface of a preform being filled and placed in an insulated chamber, and/or varying the outlet pressure. The features and capabilities of this technology are discussed below in relation to the problem of molding fairly large aircraft composite structures using various grades of resins and reinforcing fabrics. The experimental dependencies of the compressibility and permeability of preforms, thermokinetics and the rheology of resins currently obtained on samples make it possible to clarify the corresponding model dependencies. The use of these data in the described simulation tool provides a reliable prediction of the quality and performance of a real production process. The results of experiments with real full-size composite structures will be presented in our next article.

This article presents the formulation, finite element implementation and results of solving the problem of modeling two successive processes: filling a thin-walled dry compressible preform with a liquid resin, the rheology of which depends on the temperature and time, and the subsequent application of controlled pressures in three modes, differing in the magnitude and time instants of these post-infusion external pressure applications. A thin-walled structure with double curvature, reliably fixed on the surface of a form-building composite mold, is studied as an infused preform. These assumptions allow us to significantly simplify the formulation of the modeling problem, considering the preform as a shell of variable thickness with a fixed back surface. The results show that the correct consideration of the liquid resin front propagation, the thermokinetics and the viscosity of the thermosetting resin moving along the preform with changing permeability, with exothermal heat generation, allow, through a rational choice of modes, the significant improvement of such quality indicators of the vacuum infusion process. These are the uniformity of thickness distribution, fiber volume fraction and, ultimately, the accuracy of the geometry and strength properties of the molded shell-like structure.

## 2. Modeling of Sequential Vacuum Infusion and Post-Infusion Molding of Composite Preforms

The problem of modeling the process under study was solved using the example of a composite preform with transversal anisotropy of an elastic porous frame and a wall thickness of 3 mm. Its general view (see Figure 3a) shows the resin supply ports and the vacuum port. Local coordinate systems (see Figure 3b) are introduced on the front surface of the preform and are used to describe the movement of liquid resin in an anisotropic medium. The material of the forming mold is assumed to be isotropic. 

The forward problem of modeling the evolution of the infusion and subsequent post-infusion stages of the process was solved in the environment of a finite element package, Comsol Multiphysics 6.1. For the statement of the modeling problem, we mainly used the approach previously developed and presented in articles [13,16], but with significant modifications caused by the need to correctly describe the spatial anisotropy of compressibility, the permeability of the preform and the flow of a viscous fluid in a thin-walled porous body bounded by curved surfaces. The system of governing equations included the phase field Equation (1) that models the propagation of the liquid resin front along the preform:(1)∂φ/∂t+u⋅∇φ=∇⋅γ∇G,
where the dependent variable $\varphi \in \left[-1;1\right]$ determines the local resin filling *Vr* according to $V_{r}=\left(\varphi +1\right)/2\in \left[0;1\right]$, *G* is the chemical potential, *γ* is the phase mobility, and **u** is the resin superficial velocity. The initial condition for Equation (1) corresponds to the empty preform volume
(2)$$\varphi \left(0\right)=-1,$$
and the conditions at the boundaries of an impenetrable vacuum bag covering the preform, inlet and outlet are taken as (3), (4) and (5), respectively:(3)$$n^{bag}\cdot \nabla \phi \left(t\right)=0;$$
(4)$$\phi^{inl}\left(t\right)=1\Rightarrow V_{r}^{inl}=1;$$
(5)$$\phi^{out}\left(t\right)=-1\Rightarrow V_{r}^{inl}=0.$$

The heat transfer Equation (6) is defined in the body of the preform:(6)$$\rho_{pr}C_{pr}\partial T/\partial T+\nabla \cdot \left(-k_{pr}\nabla T\right)=Q_{exo}.$$

The initial value (7) for Equation (6) corresponds to a uniform temperature distribution, assumed to be 75 °C, taking into account the rheological properties of the thermosetting resin used. The same temperature is maintained in the inlet. All open surfaces of the preform and mold are subject to convective heat exchange with the air flow at temperature *T^ext^*, which in numerical experiments varied in the range of 80…90 C (see Equation (8)). The convective heat transfer coefficient *h* was assumed to be 10 W/(m^2^∙K). The condition of ideal thermal contact was assumed between the surfaces of the preform and mold.
(7)$$T(0)=T^{inl}(t)=\;75\;C,$$
(8)$$-n\cdot q=q_{0}=h(T^{ext}-T).$$

The thermophysical properties of the preform contained in this equation, mass density *ρ_pr_*, specific heat capacity *C_pr_* and thermal conductivity *k_pr_*, are determined using the mixing rule for the thermal properties of the air–resin mixture *(ρ_gr_*, *c_gr_*, $k_{gr}=k_{r}V_{r}+k_{g}(1-V_{r})$), dry preform (*ρ_f_*, *c_f_*, *k_f_*), local distributions of porosity ϕ or fiber volume fraction *V_f_* = (1 − *ϕ*) and resin filling *V_r_* [16,25]:(9)$$\rho_{pr}=\rho_{f}\cdot V_{f}+(1-V_{f})\cdot \rho_{gr},$$
(10)$$C_{pr}=\rho_{f}V_{f}\cdot c_{f}+(1-V_{f})\cdot \rho_{gr}c_{gr},$$
(11)$$k_{pr}={\left(\frac{\left(1-V_{f}\right)}{k_{r}V_{r}+k_{g}(1-V_{r})}+\frac{V_{f}}{k_{f}}\right)}^{-1}.$$

The heat transfer equation for a mold made of polymerized carbon fiber also has the form (6), but with other thermophysical properties (*ρ_m_*, *C_m_* and *k_m_*) that are independent of time and temperature.

The source term *Q_exo_* in Equation (6) represents the intensity of the exothermal heat generated per unit volume of the curing epoxy resin and depends on the cure rate ∂*α*/∂*t* according to the following relationship:(12)$$Q_{exo}=Q_{tot}\rho_{r}(1-V_{f})\cdot V_{r}\cdot \partial \alpha /\partial t,$$
where *Q_tot_* is the total amount of exothermal heat released during the curing of a unit mass of resin, and *ρ_r_* is the resin mass density.

The spatiotemporal evolution of the degree of cure α is described by the convection/diffusion/kinetics Equation (13) defined in the preform area filled with resin,
(13)$$\partial \alpha /\partial t-(\left[K\right]/\mu )\cdot \nabla p\cdot \nabla \alpha -\nabla \cdot (c_{\alpha }\nabla \alpha )=F(\alpha ,T),$$
where [*K*] is the permeability tensor of the porous preform, and the diffusion coefficient of degree of cure *c_α_* is expressed by the empirical formula
(14)$$c_{\alpha }=5\cdot 10^{-7}\frac{1+\exp\left(\left(\alpha -0.35\right)/0.25\right)}{1+\exp\left(\left(\alpha -0.85\right)/0.25\right)}\;m^{2}/s,$$
and source term in Equation (13).
(15)$$F\left(\alpha ,T\right)=A\cdot \frac{\left(w\exp\left(-E_{1}/RT\right)+\exp\left(-E_{2}/RT\right)\cdot \alpha^{m}\right)}{1+\exp\left(\left(\alpha -0.75\right)/0.25\right)}\cdot {\left(1-\alpha \right)}^{n},$$
takes into account the change in cure rate and in diffusion coefficient *c*_α_ when the reaction becomes diffusion-controlled towards the end of curing [26,27,28] (see Figure 4). In Equation (9), *R* is the universal gas constant, *T* is the Kelvin temperature, and the values of the parameters *E*_1_, *E*_2_, *A*, *w*, *m* and *n*, characterizing the thermokinetic properties of the thermosetting resin, were determined using a genetic algorithm based on the results of differential scanning calorimetry [29].

For Equation (13), the initial and boundary conditions are as follows. Initial state:(16)$$\alpha_{\Omega_{pr}}\left(0\right)=0.$$

For closed boundaries, the condition that the flux is equal to zero is accepted:(17)$$n\cdot (c_{\alpha }\nabla \alpha )=0.$$

Dirichlet condition on resin injection gates:(18)$$\alpha^{inl}\left(t\right)=\alpha_{inj}\left(T^{in}\right),$$
and the free α-flux $S_{\alpha }^{out}$ boundary condition on the vacuum vent:(19)$$S_{\alpha }^{out}=n\cdot \left(c_{\alpha }\nabla \alpha \right)\left(n\cdot u\right),$$
where **u** is the resin superficial velocity through vacuum vent and **n** is the unit normal vector to its boundary.

The fourth equation of the problem is Darcy’s equation, specified in curvilinear coordinate systems in the preform, relating the velocity **u** of the gas–fluid mixture to the pressure gradient ∇p
(20)$$\left\{\begin{array}{c} \frac{\partial }{\partial t}\left(\phi \rho_{gr}\right)+\nabla \cdot \left(\rho_{gr}u\right)=Q_{m}\\ u=-\frac{\left[K\right]}{\mu_{gr}}\nabla p \end{array}\right.,$$
where
(21)$$\rho_{gr}=\rho_{r}V_{r}+(1-V_{r})\rho_{g},$$
(22)$$\mu_{gr}=\mu_{r}V_{r}+\mu_{g}\left(1-V_{r}\right)$$
are the mass density and dynamic viscosity, respectively, of the moving gas–fluid mixture, depending on the filling of the pores with liquid resin *V_r_*, and determined by the mixing rule: *Q_m_* is the volumial source/sink of mass taken equal to zero.

For the studied transversely isotropic material of the preform, all components of the permeability tensor [*K*], which are accepted in diagonal form, satisfy the Kozeny–Carman model [29], where the minimum permeability parameter ${\tilde{k}}_{n}$ normal to the preform surface is taken to be 4 times less than the in-plane minimum permeability accepted as ${\tilde{k}}_{t}=4\cdot {\tilde{k}}_{n}=2\cdot 10^{-10}\;m^{2}$. The value of the fiber volume fraction *V_f_*, which significantly affects the properties of the preform in accordance with Formulas (9)–(11), is a function of compressive pressure, equal to the difference between the pressure applied to the preform from the outside and the pore pressure, and also depends on the filling of the pores in the preform with liquid resin *V_r_*. Our study uses the rule of mixtures to quantitatively describe this relationship:(23)$$V_{f}=1-\phi =V_{r}\cdot V_{f}^{wet}+\left(1-V_{r}\right)\cdot V_{f}^{dry},$$
where the dependencies of $V_{f}^{dry}$ and $V_{f}^{wet}$ on compressive pressure $p_{comp}(r)=p^{appl}-p(r)$ (see Figure 5) are described by similar empirical formulas, differing only in the values of the coefficients:(24)$$V_{f}^{dry,wet}=V_{f\_\min}^{dry,wet}+b^{dry,wet}\cdot a\cosh\left(1+p_{comp}/a^{dry,wet}\right).$$

Correct accounting of the dependence of resin viscosity $\mu_{r}\left(T,\alpha ,t\right)$ on time, the degree of cure α(T,t) and temperature *T* is provided by the empirical model (25), which is a satisfactory approximation of experimental data in the used ranges of α(T,t) and *T*. The coefficients $\upsilon_{1},\upsilon_{2}$ are determined from the results of viscometry.
(25)$$\mu_{r}\left(T,\alpha ,t\right)=\mu_{r}^{0}(T^{in})\cdot \exp\left(\upsilon_{1}\cdot (T(t)-T^{in})+\upsilon_{2}\cdot \alpha (T,t)\right).$$

At the initial time instant, the pore pressure *p*(0) in the preform and the outlet pressure are equal to the vacuum pressure taken as 20 kPa. The pressure applied to the external surface of the preform *p^appl^*(0) is equal to atmospheric *p^atm^* = 100 kPa.
(26)$$p(0)=p^{out}(0)=p^{vac}=\;20\;kPa,$$
(27)$$p^{appl}(0)=p^{inl}(0)=p^{atm}=\;100\;kPa.$$

Below are the results of modeling four process control scenarios (Modes 1–4) at the post-infusion stage, which always begins from the moment the minimum level of resin filling of the preform is decreased to a value of 0.1. The duration of the simulated process was assumed to be 6 h for all studied modes. Modes 1–3 are implemented with open inlets through which the resin is supplied under atmospheric pressure, and in Mode 4 the injection gates are blocked at the beginning of the post-infusion stage of the process. All these modes are presented in Figure 6 in the form of time dependencies of applied *p^appl^*(*t*), inlets *p^inl^*(*t*) and outlet *p^out^*(*t*) pressures. In these diagrams, the triangular icons on the time axis indicate a start–stop raise in external pressure 
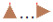
 and vacuum vent pressure 
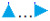
, respectively.

Mode 1 is a conventional vacuum infusion without changing any controlled pressures.

Mode 2 includes, at the post-infusion stage, a gradual increase in pressure in the vacuum port from vacuum to atmospheric in order to equalize the pore pressure and thickness of the preforms,
(28)$$p^{out}(t)=p_{in}^{out}+(p^{atm}-p_{in}^{out})\cdot H\left(t_{start}^{out},\Delta t_{start},t\right),$$
where $t_{start}^{out}$ is the moment the pressure rise begins, $\Delta t_{start}$ is the duration of the pressure rise to the final value, and *H* is the smoothed Heaviside function. It is important to note that the moment $t_{start}^{out}$ at which the outlet pressure begins to increase is selected at the final stage of the process, when the viscosity of the resin has significantly increased, but its hardening has not yet occurred. This is necessary to prevent air from being introduced back into the preform through the outlet.

Mode 3 differs from Mode 2 in that the increase in outlet pressure is preceded by an increase in external pressure compressing the preform,
(29)$$p^{appl}(t)=p^{atm}+p^{add}\cdot H\left(t_{start}^{appl},\Delta t_{start},t\right),$$
where the time $t_{start}^{appl}$ of the compressive pressure application differs from the moment of increase in outlet pressure by approximately an hour, as is customary in this work: $t_{lag}=t_{start}^{out}-t_{start}^{appl}≅60$ min. The results presented below correspond to an additional compressive pressure $p^{add}$ = 80 kPa.

Mode 4 differs from Mode 3 only in that at moment $t_{start}^{appl}$ both injection ports are closed, eliminating the flow of resin into the preform.

All parameters of the mold material and resin included in Equations (1)–(25) were determined similarly to our previous works [13,16].

After eliminating holes and curved joints of surface patches that disimprove the mesh, the geometry of the preform assembled with the mold was imported into the finite element package Comsol Multiphysics 6.1. Following the separation of the areas of the injection and vacuum ports, a FE mesh was constructed, as shown in Figure 7.

The simulated process time for all modes studied was set to 6 h, when the resin was intensively polymerized. The simulation of the transient process in Modes 1–3 was carried out in one stage, during which only the values of the controlled pressures could change. In contrast, the simulation of Mode 4 was carried out in two stages. The second stage began immediately before the application of external pressure at the moment $t_{start}^{appl}-\Delta t_{start}$ when the boundary condition at the injection ports changed to blocking.

Two significant features of the system under consideration significantly reduce the computational complexity of its modeling, abandoning the solution to the problem of mechanics of a three-dimensional porous deformable body. This is the thin-walledness of the infused preform, which allows it to be considered as a shell [30], compressible in thickness, and the fixation of the inner surface of this shell on a non-deformable mold. The second circumstance allows us to neglect the deformation of the preform in the tangent plane to its surface. Thus, the deformation of the preform in the direction normal to its surface should be studied as one-dimensional.

An experimental study of the preform material compressibility, the results of which are presented in Figure 5, and empirical dependencies (24), make it possible to use these dependencies when operating a computer model of the process. It allows a determination of the local fiber volume fraction *V_f_* on each simulation time step from the calculated values of pore pressure *p* and external *p^appl^*. Then, the dependence of the preform thickness *h* on *V_f_* is determined from the simple relation
(30)$$h=h_{in}\cdot V_{f,in}/V_{f},$$
where *h_in_* and *V_f,in_* are the initial thickness and the fiber volume fraction of the uncompressed preform, respectively.

For a detailed analysis of the evolution of process parameters during simulation and a presentation of the results in a visual form, the probes of the minimum, maximum and average values of the parameters of interest and their integrals over these areas were defined in the preform volume and on its surfaces.

## 3. Results and Discussion

All numerical experiments, the results of which are presented below, were performed with an unchanged thermosetting resin and reinforcing matrix of the molded composite structure. This is due to the fact that the purpose of our study is only to identify the features of each of the four modes under consideration, which should be taken into account when choosing an optimization strategy. When producing thin-walled composite structures using vacuum-infusion technology, it is necessary to achieve the maximum value of the fiber volume fraction with its minimum variation in the body of the structure. This will also result in minimizing the preform wall thickness and its variations. When choosing controlled parameter values for each of the scenarios under study, it is necessary to eliminate the occurrence of undesirable situations, such as the reverse penetration of air into the resin-filled preform through the vacuum port, and also, if possible, reduce the duration of the process until the resin is completely cured.

To understand the mechanics of the processes occurring in each of the molding modes, the most informative is the dependence of pore pressure on the modes. The importance of this characteristic of the processes is due to the fact that pore pressure *p* has a decisive influence on the fiber volume fraction *V_f_* (see Equation (24)), and that, in turn, on the wall thickness *h* (30). Figure 8 shows the time dependencies of the average pore pressure over the preform and its average deviation. Until the time point of 105 min, when the increase in external compressive pressure started, all four dependencies coincided. In addition, all the time dependencies and 3D screenshots presented below for Modes 1 and 2 coincided up to the moment of 165 min, when the pressure in the vacuum port began to increase. The general difference between controlled Modes 2, 3 and 4 from the uncontrolled vacuum infusion (Mode 1) is that by the time of 240 min, the average pressure <*p*> in the preform in these modes had reached a constant value of 100 kPa, and its variation <|δ*p*|> tended to zero, which indicates a uniformity of pore pressure distribution in the preform. On the other hand, both the average pore pressure and its variation in uncontrolled Mode 1 changed significantly until the end of the simulated process duration.

The dependencies presented in Figure 8 demonstrate that the change in pore pressure is always somewhat delayed after a change in pressure both on the outer surface of the preform covered with a vacuum bag and in the vacuum port, although the duration of the change in controlled pressure is quite significant, amounting to 30 min. Obviously, this inertia depends on the viscosity, that is, the fluidity of the resin and the permeability of the preform. With higher viscosity and higher compression pressure, this inertia will increase, which should be taken into account when assigning moments of change in controlled pressures.

A more detailed idea of the processes occurring in the preform during the entire infusion stages is given by Figure 9, Figure 10, Figure 11 and Figure 12, which present screenshots combining images of pore pressure distribution, resin fill levels and streamlines indicating the directions of liquid resin flow. The size of the arrowheads is proportional to the resin superficial velocity in the indicated direction.

Figure 10c,d, Figure 11f and Figure 12b,d clearly show that the consequence of increasing pressure in the vacuum port is a slowing down and then a complete stop of the movement of the resin at some point in time. This increase in pressure in the vacuum line, necessary at the final stage of the process, when the resin has begun to harden but still retains some fluidity, makes it possible to equalize the pore pressure in the preform and, consequently, the distribution of *V_f_* and *h* before the complete hardening of the resin. A certain increase in the average pore pressure at the moment of external compression application, and then its decrease and settling, is due to the redistribution between the pressures of the liquid resin and those in the elastic porous frame of the preform, the sum of which counteracts the applied outside pressure.

Figure 13 shows the fluid resin average velocity module |**u**| for all studied control modes. It should be noted that this velocity is not the velocity of the resin flow towards the vacuum port, but is an indirect characteristic describing the average mobility of the resin at each time. However, it predicts with a sufficient degree of reliability the moment when the resin flow stops. From the graphs in Figure 13, it can be seen that all controlled modes shorten the period of intensive resin movement, but the resin movement stops first in Mode 3. This result suggests that this mode may provide the best performance. However, it is necessary to ensure that the preform is completely filled with resin before gelation, which suggests the advisability of a later application of controlled pressures.

A visual representation of the dynamics of filling the preform with resin is given by the time diagrams of the mass flows of the resin–air mixture passing through the injection gates and vacuum vent, shown in Figure 14. Diagrams in Figure 14b show that at the final stage of the process, when the preform is almost completely filled with resin and the controlled pressures are constant, the mass flows through the injection and vacuum ports are the same, which confirms the correct description of the physics of the simulated processes. In addition, the flow of resin in Mode 1 becomes steady, and the flow in Modes 2–4 stops almost simultaneously. Thus, non-zero values of resin velocities in Modes 2–4 after ~200 min (see Figure 13) are caused by some of its movements inside the preform, and not directed towards the outlet (see Figure 10, Figure 11 and Figure 12).

Meanwhile, the most important information about the achievable quality of the process is provided by the dependence of the fiber volume fraction and the wall thickness of the molded preform. These diagrams, presented in Figure 15 and Figure 16, are constructed using the initial values of the *V_f,in_* = 0.45 and thickness *h_in_* = 3.05 mm experimentally measured in the eight-layer carbon fiber preforms under study. A comparative analysis of the diagrams for *V_f_* shows that the best results are demonstrated by Mode 3: <*V_f_*> = 0.58; max(|δ*V_f_*|) = 0.027. Modes 3 and 4, which use the application of external compressive pressure and provide the highest value of <*V_f_*> compared to Modes 1 and 2, even before the pressure in the vacuum port is equalized, lead to a significant homogenization of <*V_f_*> in the preform volume. This also allows us to predict better reliability and repeatability of the molding process results with Modes 3 and 4. Despite the best *V_f_* uniformity in Mode 2, the low value of <*V_f_*> = 0.51 makes it unsuitable for use in the production of critical composite structures. The customary vacuum infusion in Mode 1 showed the worst result.

The functional relationship (20) between *V_f_* and *h* leads to the conclusion that such indicators of process quality as the thickness and homogeneity of the preform thickness will also be better for the process implemented in Mode 3 (see Figure 17).

The last two figures show that the final time interval between 240 and 360 min is characterized by a practically unchanged average value of *V_f_* in controlled modes 2–4, which is explained by the slowing down and stopping of the resin flow. During this time, intensive polymerization of the resin occurs and its viscosity increases. This can lead to a critical situation in which the area of the preform around the outlet that is not completely filled with resin is blocked from resin flowing out. This results in an unacceptable increase in porosity around the outlet of the finished part. The state of the output port at the final stage of the process is shown in Figure 18 for all studied modes. The viscosity distribution in the preform at the final stages of the process implemented in mode 3 is shown in Figure 19.

Obviously, the best situation at the final stage of the process should be one in which the level of resin filling of the exit port $V_{r}^{out}$ is maximum, and its viscosity around the outlet $\mu_{r}^{out}$ is minimal. At the same time, this viscosity $\mu_{r}^{out}$ should approach the onset of solidification in order to prevent air from entering the preform when the pressure at the outlet port increases. Such a rational choice of process modes should ensure the stability and repeatability of quality indicators. It is important to note that although the resin fill level at the outlet port in Mode 3 is less than in Modes 1 and 2 (see Figure 18 and Figure 19), Mode 3 provides significantly better fiber volume fraction levels. When discussing the influence of process modes on the nature of the resin flow (see Figure 14), it was found that changing the control pressures of $p^{appl}\left(t\right)$ or $p^{out}\left(t\right)$ always led to a slowing down and complete stop of the resin flow. This allows us to conclude that a slight shift in the beginning of the growth of these pressures towards later times can significantly increase the filling of the vacuum port zone with resin at the final stage of the process. Such a shift cannot worsen the stability of the process due to an increase in the viscosity of the resin at the outlet, since the result of some acceleration of its flow will be the entry of a resin with a lower degree of cure into the outlet zone. The result of stopping the resin later is a slight widening of the cure rate curve towards the later time, as shown in Figure 20.

## 4. Conclusions

The numerical experiments carried out in comparison with the experimental data presented in works (28,29) showed that the finite element modeling tool developed makes it possible to predict with a high degree of reliability the behavior and results of the considered types of vacuum-infusion processes, including those with a post-infusion stage, under which the boundary conditions at the injection and vacuum ports can be changed. Our simulations results confirmed the effectiveness of using controlled pressures at the post-infusion stage of the process. Using the correct strategy to control the external pressure applied to the open surface of the preform and the pressure in the vacuum line allows you to obtain the maximum achievable values of the fiber volume fraction and the uniformity of its distribution in the molded composite part. The dependence of the speed of liquid resin propagation in a compressible porous preform on temperature and the pressure gradient causing the movement of the resin, as well as permeability, which depends on the compressive pressure, make it possible to control the performance of vacuum infusion processes by regulating temperature, external and vacuum pressures. This method of molding composite structures can be implemented in special chambers isolated from the external atmosphere, or in autoclaves with a vacuum line supplied to the preform.

Vacuum infusion processes, as well as resin transfer molding, are also successfully modeled using the presented software, which requires the entire set of properties of the molded preform components and the forming equipment. The use of CAD model geometry allows you to simulate and optimize the processes of forming composite parts of arbitrary complexity and size. This conclusion is confirmed by the quite acceptable duration of simulation of each of the above considered process modes, which is 40–55 min on an average-performance computer. The main direction of our further research is related to the use of developed numerical and experimental methods in the manufacture of a wide range of aircraft structures of various sizes and complexity. Our next publications are devoted to the results of applying the proposed methodology, confirmed by full-scale experiments.

## Figures and Tables

**Figure 1 polymers-16-00654-f001:**
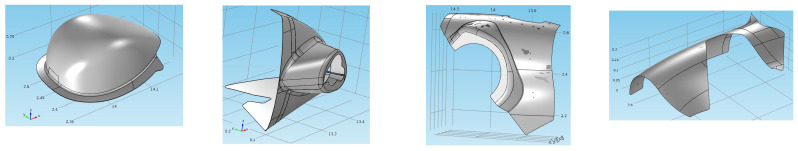
Examples of thin-walled composite structures produced using vacuum infusion technology.

**Figure 2 polymers-16-00654-f002:**
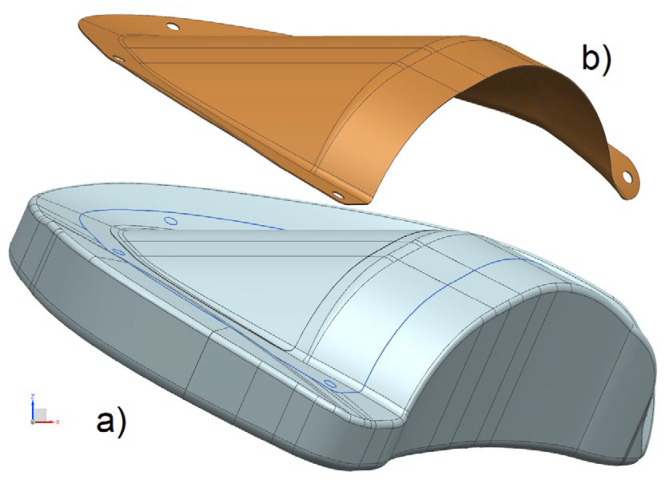
CAD models of a forming open mold (**a**) and a thin-walled composite structure produced on it (**b**) using vacuum infusion technology.

**Figure 3 polymers-16-00654-f003:**
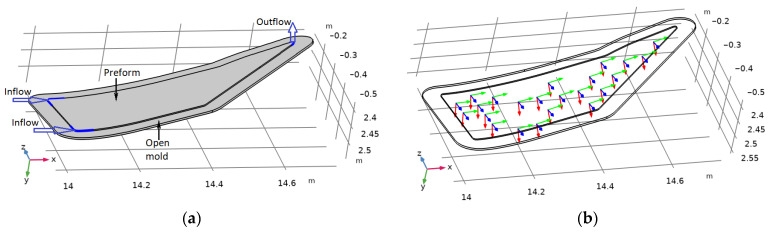
(**a**) Geometry of the modeled system. (**b**) Local coordinate systems defined on the face surface of the preform.

**Figure 4 polymers-16-00654-f004:**
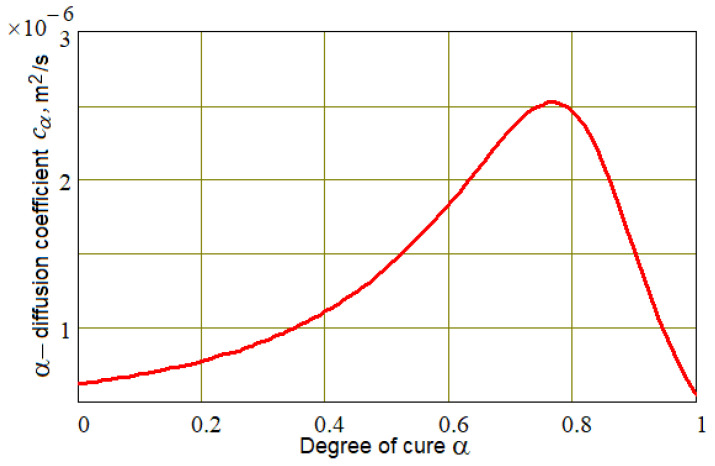
Empirical dependence of the diffusion coefficient *c_α_* on the degree of cure α.

**Figure 5 polymers-16-00654-f005:**
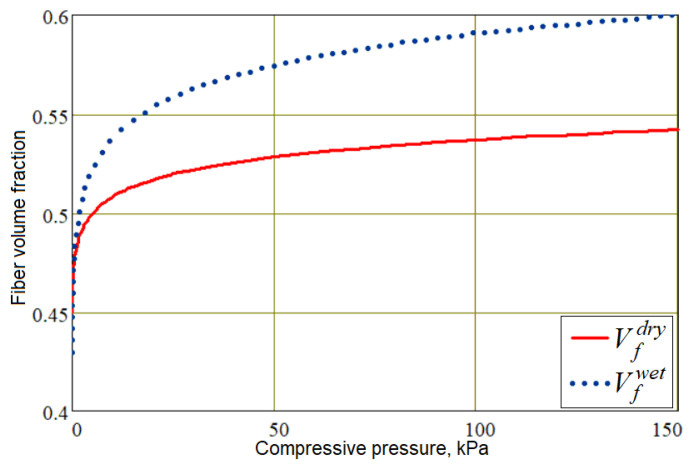
Dependencies of the preform compaction in dry and wet states [16].

**Figure 6 polymers-16-00654-f006:**
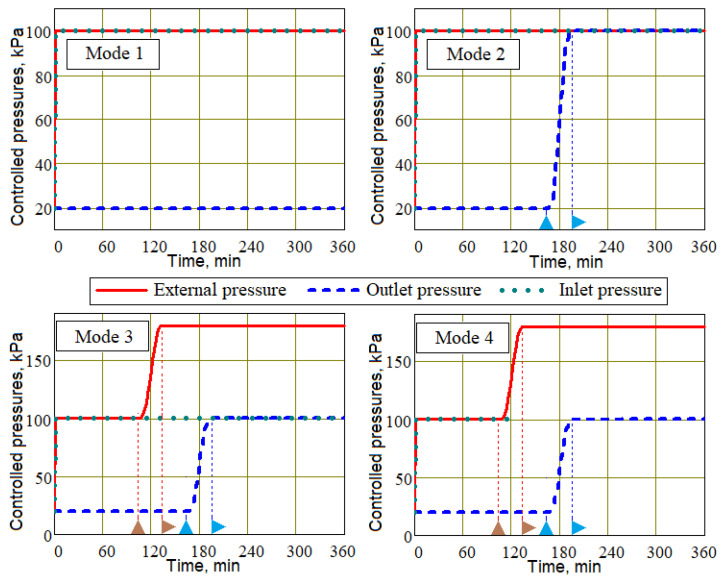
Time dependencies of the controlled pressures for process Modes 1–4.

**Figure 7 polymers-16-00654-f007:**
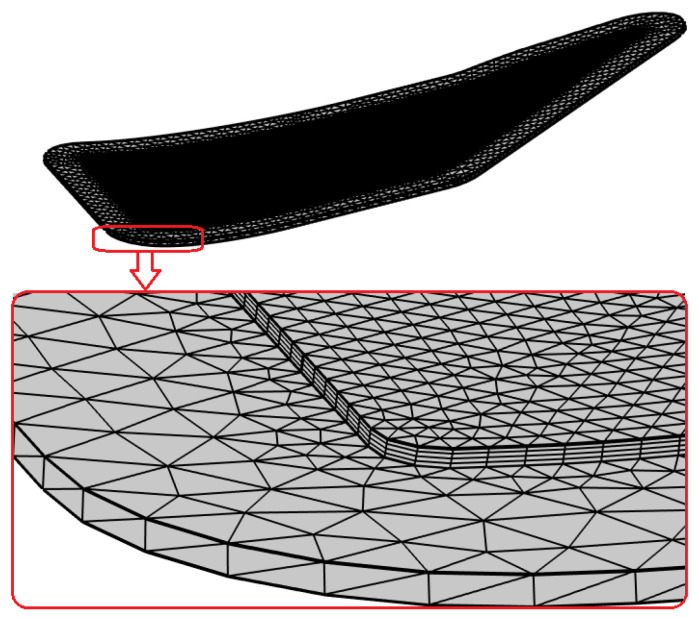
The FE model of the assembled mold and preform after meshing.

**Figure 8 polymers-16-00654-f008:**
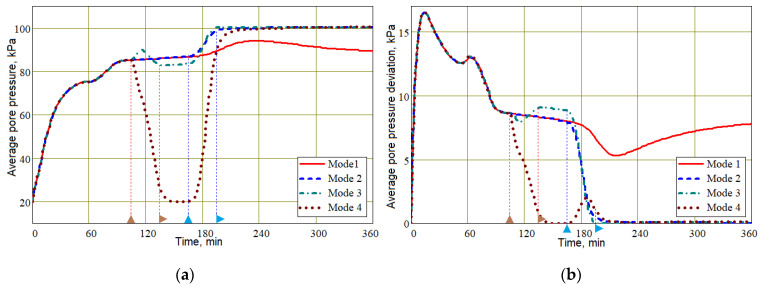
Time dependencies of the average pore pressure in the infused preform (**a**) and its average variation (**b**) for the four studied modes of the vacuum-infusion process.

**Figure 9 polymers-16-00654-f009:**
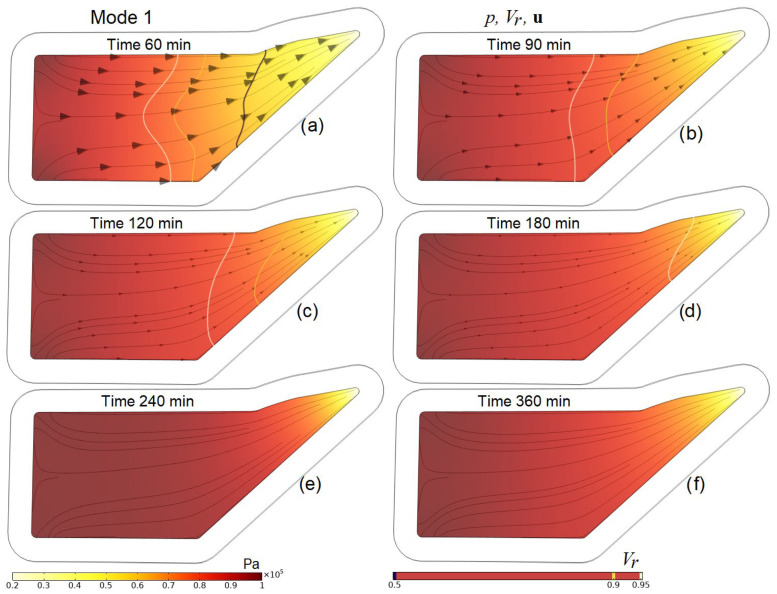
Screenshots depicting the distribution of pore pressure *p* in the preform, the levels of its filling with resin *V_r_* and streamlines **u** at characteristic times in the process implemented in Mode 1 (uncontrolled vacuum infusion).

**Figure 10 polymers-16-00654-f010:**
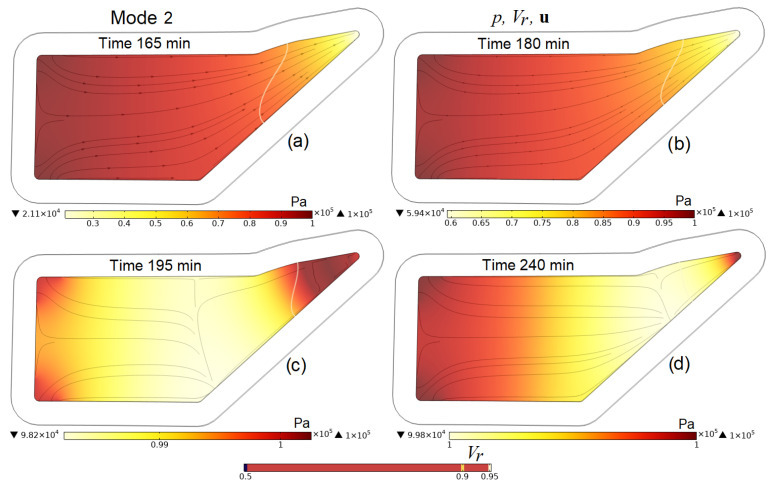
Screenshots depicting the distribution of pore pressure *p* in the preform, the levels of its filling with resin *V_r_* and streamlines **u** at characteristic times in the process implemented in Mode 2 (gradual increase in pressure in the vacuum port from $p_{in}^{out}$ to $p^{atm}$ from 165 to 195 min).

**Figure 11 polymers-16-00654-f011:**
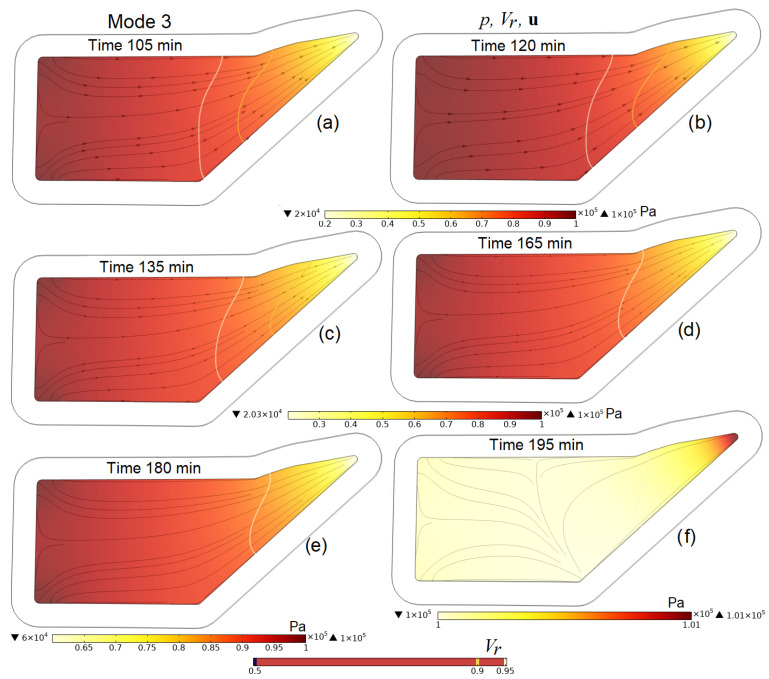
Screenshots depicting the distribution of pore pressure *p* in the preform, the levels of its filling with resin *V_r_* and streamlines **u** at characteristic times in the process implemented in Mode 3 (gradual increase in pressure on the outer surface of the preform covered with a vacuum bag from $p^{atm}$ to $p^{atm}+p^{add}$ from 105 to 135 min and in the vacuum port from $p_{in}^{out}$ to $p^{atm}$ from 165 to 195 min).

**Figure 12 polymers-16-00654-f012:**
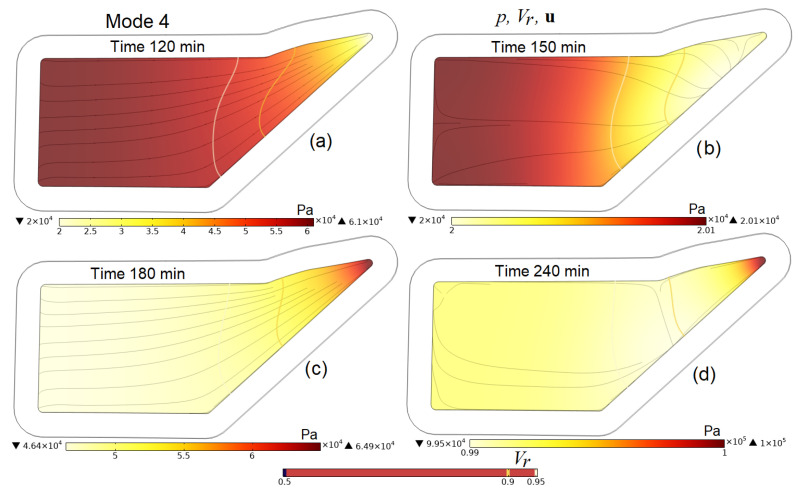
Screenshots depicting the distribution of pore pressure *p* in the preform, the levels of its filling with resin *V_r_* and streamlines **u** at characteristic times in the process implemented in Mode 4 (simultaneous closing of both resin gates at 120 min and gradual increase in pressure in the vacuum vent from $p_{in}^{out}$ to $p^{atm}$ from 165 to 195 min).

**Figure 13 polymers-16-00654-f013:**
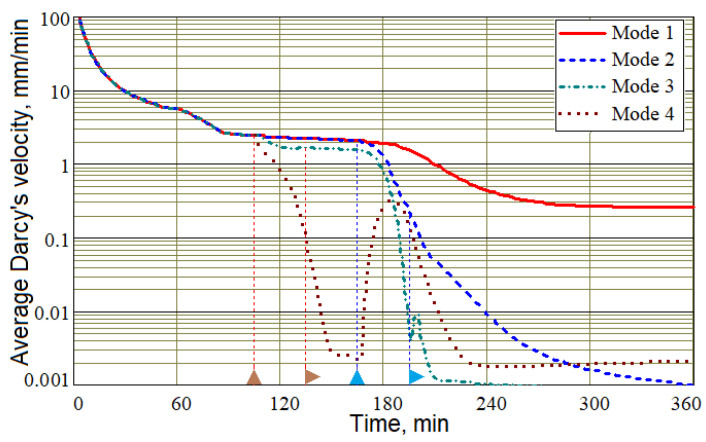
Time histories of the fluid resin average velocity module |**u**| for control Modes 1–4.

**Figure 14 polymers-16-00654-f014:**
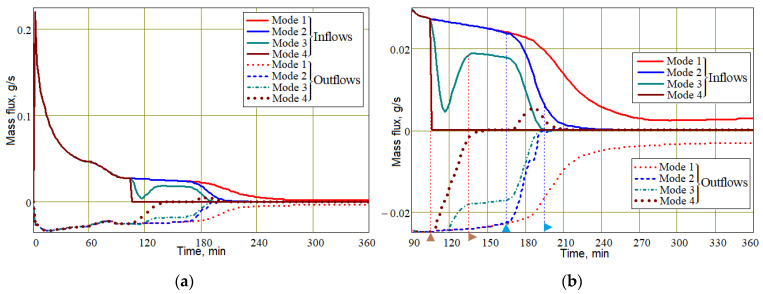
Time dependencies of the inlets and outlet mass fluxes for the studied control modes: (**a**) throughout the entire vacuum infusion process and (**b**) at the final stage of the process under the action of control pressures.

**Figure 15 polymers-16-00654-f015:**
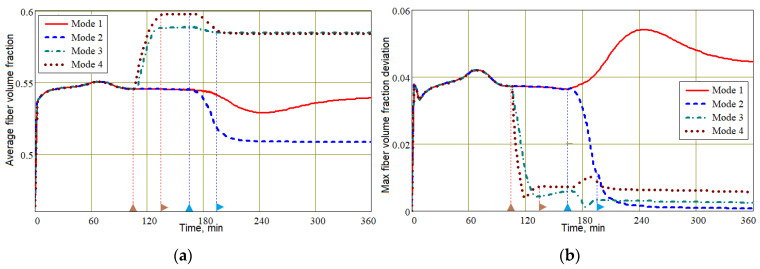
Time dependencies of the average fiber volume fraction <*V_f_*> (**a**) and its maximum deviation max(|δ*V_f_*|) (**b**).

**Figure 16 polymers-16-00654-f016:**
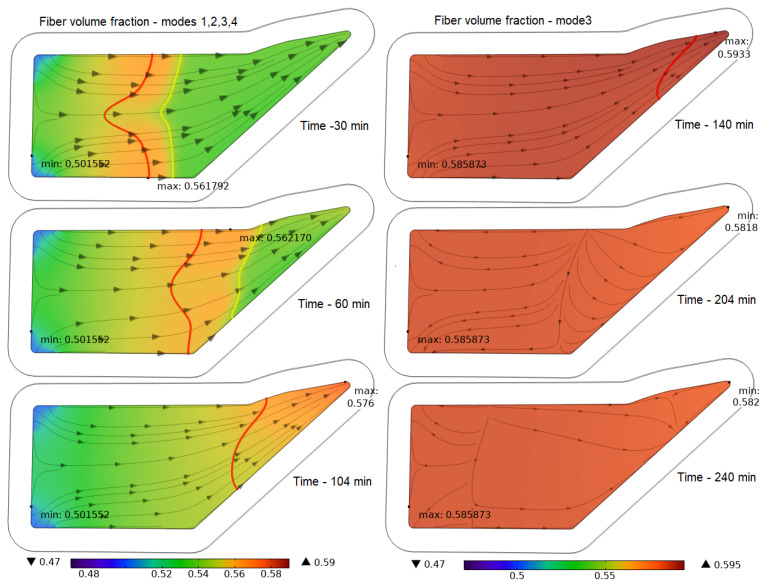
Screenshots depicting the combined distribution of the fiber volume fraction *V_f_*, levels 0.5 (yellow) and 0.9 (red) of preform filling with resin *V_r_* and streamlines **u** at characteristic times in the process implemented in Mode 3.

**Figure 17 polymers-16-00654-f017:**
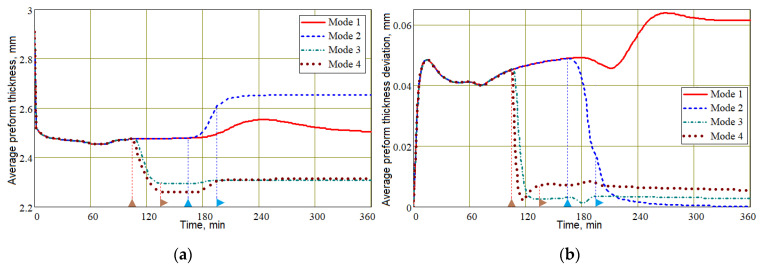
Evolution of the average preform thickness (**a**) and its deviation (**b**) during the vacuum infusion process for all studied modes.

**Figure 18 polymers-16-00654-f018:**
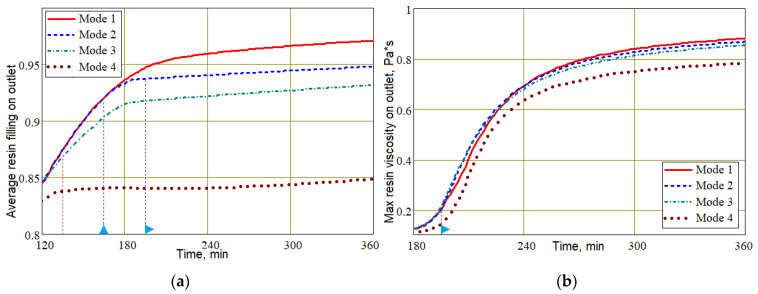
The average filling with resin (**a**) and maximum resin viscosity (**b**) around the outlet.

**Figure 19 polymers-16-00654-f019:**
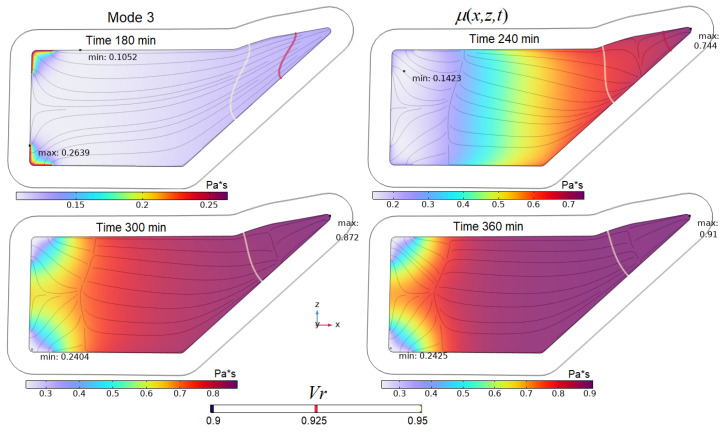
Screenshots showing the joint distribution of resin viscosity *μ_r_*, resin fill levels *V_r_* and streamlines **u** at the final stages of the process implemented in Mode 3.

**Figure 20 polymers-16-00654-f020:**
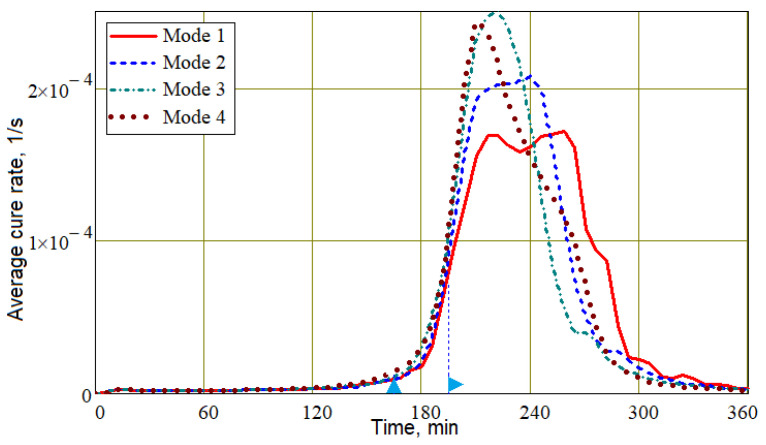
The time dependencies of the average cure rate in the preform for the Modes 1–4.

## Data Availability

Data are contained within the article.

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
