# Peer review of "Numerical Study of Thin-Walled Polymer Composite Part Quality When Manufactured Using Vacuum Infusion with Various External Pressure Controls"

_polymers, 2024, doi:10.3390/polym16050654_

Round 1
Reviewer 1 Report
Comments and Suggestions for Authors
This paper presents the formulation, finite element implementation and results of solving the problem of modeling two successive processes: filling a thin-walled dry compressible preform with a liquid resin, the rheology of depending on the temperature and time, and subsequent application of controlled pressures in three modes, differing in the magnitude and time instants of these post-infusion external pressures application.
Comment:
The numerical simulation is conducted. However, there is no experimental validation. The related experiment should be conducted to validate the simulation results.
Author Response
Dear reviewers!
We are very grateful for your attention and advices, which we hope will improve the content and readability of our article.
Our responses to your comments and the corresponding changes in the text of the article are highlighted.

Reviewer 2 Report
Comments and Suggestions for Authors
The paper showcases findings from simulating diverse methods for vacuum infusion molding thin-walled polymer-composite structures with flexible geometries. Leveraging the minimal thickness of the produced structures and their secure attachment to the mold's inflexible surface, the process model was notably streamlined. This model effectively incorporates the dynamics of thermosetting resin flow, considering its evolving rheological properties within a compressible porous preform of intricate 3D shapes. Additionally, it addresses alterations in boundary conditions at both injection and vacuum ports throughout the post-infusion molding phase. The work is significant for thin-walled polymer composites; however, it has some minor flaws commented below:
Please write the initial conditions for all governing equations (1), (2), (7) and (10), as well as boundary conditions for the convection/diffusion/kinetics equation (7) and for the equation Darcy (10).
Author Response

(The authors gave the same response as above.)

Reviewer 3 Report
Comments and Suggestions for Authors
The manuscript entitled "Numerical study of the thin-walled polymer composite parts quality manufactured using the vacuum infusion with various external pressure controls" is a valid research work with appropriate level of novelty and originality. The topic of the manuscript fits well the scope of the journal. The scientific and technical quality of the work is appropriate. The introduction section clearly represents the state of the art in the specific field of the research, which is well substantiated by the relevant references including the recent ones. The obtained results are described and treated mostly correctly. The conclusion section clearly summarizes the presented results. The results presented in the manuscript should have definite impact on the specific field of the research. However, authors should address several issues before the manuscript can be considered for acceptance. Here are the specific comments:
(A) This work is based on the modelling. Are there any confirmation of the proof-of-concept based on the experiments? If so, please add the relevant information to the manuscript.
(B) The first two sentences of the conclusion section should be fully skipped being irrelevant.
(C) The style of the language should be checked by the native speaker; this will increase the overall clarity of the manuscript.
Comments on the Quality of English LanguageThe style of the language should be checked by the native speaker; this will increase the overall clarity of the manuscript. Some sentences are too long to be clearly understandable.
Author Response

(The authors gave the same response as above.)
